# Reasoning Like Humans: Enhancing Multi-Image Reasoning Via Cognition-Inspired Meta-Action Framework

## Abstract

While Multimodal Large Language Models (MLLMs) excel at single-image understanding, they exhibit significantly degraded performance in multi-image reasoning scenarios. Multi-image reasoning presents fundamental challenges including complex inter-relationships between images and scattered critical information across image sets. Inspired by human cognitive processes, we propose the Cognition-Inspired Meta-Action Framework (CINEMA), a novel approach that decomposes multi-image reasoning into five structured meta-actions: Global, Focus, Hint, Think, and Answer which explicitly modeling the sequential cognitive steps humans naturally employ. For cold-start training, we introduce a Retrieval-Based Tree Sampling strategy that generates high-quality meta-action trajectories to bootstrap the model with reasoning patterns. During reinforcement learning, we adopt a two-stage paradigm: an exploration phase with Diversity-Preserving Policy Optimization (DiPO) to avoid entropy collapse, followed by an annealed exploitation phase with DAPO to to gradually strengthen exploitation. To train our model, We construct a dataset of 57k cold-start and 58k reinforcement learning instances spanning multi-image, multi-frame, and single-image tasks. We conduct extensive evaluations on multi-image reasoning benchmarks, video understanding benchmarks, and single-image benchmarks, achieving competitive state-of-the-art performance on several key benchmarks. Our model surpasses GPT-4o on the MUIR and MVMath benchmarks and notably outperforms specialized video reasoning models on video understanding benchmarks, demonstrating the effectiveness and generalizability of our human cognition-inspired reasoning framework.

## 1 Introduction

Recent Multimodal Large Language Models (MLLMs) have demonstrated remarkable capabilities in single-image understanding tasks (Bai et al., 2025; Chen et al., 2024b; Hurst et al., 2024; Li et al., 2024b; Wang et al., 2024a), with extensive research focusing on enhancing models' single-image reasoning abilities (Wang et al., 2025e; Huang et al., 2025; Chen et al., 2025a; Yang et al., 2025b). However, real-world applications often involve processing multiple images simultaneously, such as in e-commerce, autonomous driving, and video content understanding. Despite their success in single-image tasks, MLLMs exhibit significantly degraded performance when handling multi-image reasoning scenarios (Wang et al., 2025a; Meng et al., 2025b).

Multi-image reasoning presents two fundamental challenges. First, images often exhibit complex inter-relationships: semantic associations, spatial arrangements, temporal sequences, that are crucial for task completion yet require sophisticated integration beyond isolated image processing (Zhang et al., 2025b; Meng et al., 2025b). Second, critical information may be scattered across specific images within larger sets, demanding precise identification and focus on relevant visual content while filtering out distractors.

Human cognition provides valuable insights for addressing these challenges. When faced with complex multi-image reasoning tasks, humans typically employ a systematic approach: they first survey the entire problem to understand its global structure, then focus on key relevant details, identify potential pitfalls and confusing elements, engage in deliberate reasoning to connect information across

images, and finally synthesize their analysis into a coherent solution. This natural cognitive process suggests that artificial reasoning systems would benefit from structured meta-cognitive frameworks that explicitly model these human-like reasoning patterns.

Motivated by these observations, we propose the **C**ognition-**In**spired **M**eta-**A**ction Framework (CINEMA) that addresses multi-image reasoning through three key innovations. First, we introduce a set of five meta actions: Global, Focus, Hint, Think, and Answer, which systematically guide models through human-inspired reasoning processes. These meta actions provide a structured cognitive framework that enables models to effectively navigate the complexities of multi-image reasoning by explicitly modeling the sequential cognitive steps that humans naturally employ. Second, we develop a Retrieval-Based Tree Sampling strategy that mirrors human learning dynamics through a student-teacher paradigm. This approach generates diverse, high-quality reasoning trajectories by first allowing a student model to attempt initial solutions, then having a teacher model refine these attempts, and finally retrieving alternative solution paths from a database of reasoning trajectories. process not only ensures the quality of training data but also refines the reasoning trajectories, enabling the model to generate reasoning patterns that more closely resemble human-like thinking. Third, we design a novel two-stage reinforcement learning approach to optimize the reasoning process while maintaining trajectory diversity. We observe that standard reinforcement learning often suffers from entropy collapse (Wang et al., 2025d; Cui et al., 2025; Li et al., 2025), where policies become overly deterministic and lose exploration capacity over time. To address this challenge, our first stage employs **Di**versity-**P**reserving Policy **O**ptimization (DiPO) with a trajectory homogeneity penalty to maintain sufficient exploration and prevent premature convergence to suboptimal solutions. The second stage then applies dynamic sampling policy optimization (DAPO) (Yu et al., 2025) to gradually transition toward more focused behaviors, effectively balancing the exploration-exploitation trade-off throughout the training process.

To train our model, we construct a high-quality training dataset comprising 57k cold-start instances and 58k reinforcement learning instances. Each cold-start instance contains two distinct reasoning trajectories to provide diverse supervision signals during initial training. The dataset encompasses three categories of visual reasoning tasks: multi-image tasks, multi-frame tasks and single-image tasks. Our main contributions are as follows:

- We propose a human cognition-inspired reasoning framework that decomposes complex multi-image reasoning into five structured meta actions (Global, Focus, Hint, Think, Answer). This framework systematically models the sequential cognitive processes that humans naturally employ when solving multi-image reasoning tasks, providing explicit guidance for models to navigate complex visual reasoning scenarios.

- We introduce a novel Retrieval-Based Tree Sampling strategy that generates diverse, high-quality training trajectories through student-teacher interactions, coupled with a two-stage reinforcement learning paradigm: Diversity-Preserving Policy Optimization (DiPO) with trajectory homogeneity penalty to maintain exploration, followed by DAPO to consolidate performance while preserving learned diversity.

- We construct a comprehensive training dataset with 58k cold-start instances where each contains two reasoning trajectories, and 58k reinforcement learning instances across multi-image, multi-frame, and single-image tasks.

- We conduct extensive evaluations across multiple benchmarks spanning multi-image reasoning, video understanding, and single-image tasks. Our method achieves state-of-the-art performance on numerous benchmarks and notably outperforms specialized video reasoning models on video understanding tasks, demonstrating the effectiveness and generalizability of our approach.

## 2 RELATED WORK

**Multimodal Reasoning.** Recent works have enhanced MLLM reasoning capabilities (Huang et al., 2025; Dong et al., 2025; Hu et al., 2024; Su et al., 2025; Yang et al., 2025a), but most focus on single-image scenarios. Real-world applications like autonomous driving and video understanding require multi-image reasoning. Existing multi-image approaches have key limitations. Zhang et al. (2025b) propose a Focus-Centric Visual Chain that decomposes multi-image tasks into sequential sub-questions targeting specific visual subsets. However, their reasoning process mainly

focuses on individual image subsets instead of leveraging global multi-image context.. MIA-DPO Liu et al. (2025c) augments single-image datasets with unrelated images for preference optimization, but primarily handles cases where questions involve only single images within multi-image contexts. Authentic multi-image reasoning requires models to analyze individual images while comprehending holistic relationships among all images. Inspired by human cognition, we propose a reasoning framework that effectively navigates both local image analysis and global inter-image relationships.

**Reinforcement Learning for Reasoning and Entropy Control.** Early reinforcement learning approaches for foundation models relied on Reinforcement Learning from Human Feedback (RLHF), which required training a separate reward model and extensive human-labeled preference data (Ouyang et al., 2022; Hunter, 2004). Direct Preference Optimization (DPO) (Rafailov et al., 2023) simplified this pipeline but still depended on preference annotations. More recently, large-scale pure RL methods have shown strong gains in reasoning, with outcome-based rewards alone proving effective (Guo et al., 2025; Team et al., 2025; Zeng et al., 2025; Hu et al., 2025a; Liu et al., 2025b; Yan et al., 2025; Chen et al., 2025b). To regulate exploration, many approaches add entropy or KL regularization (He et al., 2025; Liu et al., 2025a), introduce entropy bonuses through reward shaping (Cheng et al., 2025), or apply stabilizing heuristics such as loss reweighting (Wang et al., 2025d; Cui et al., 2025) and clip-higher mechanisms (Yu et al., 2025). While these methods focus on entropy within a single response, others encourage diversity across responses, e.g., via embedding-based distance measures (Chen et al., 2025d) or enforcing dissimilarity in generated answers (Chen et al., 2025c). Our approach builds on this line of work but emphasizes diversity at the meta-action level for entropy control.

## 3 METHOD

The famework of our method is shown in Figure 1. We first define five structured meta actions—Global, Focus, Hint, Think, and Answer—that model human cognitive processes (Section 3.1). We then propose Retrieval-Based Tree Sampling to generate diverse, high-quality training trajectories via student-teacher interactions (Section 3.2), and construct a comprehensive dataset with 58k cold-start and 58k reinforcement learning instances (Section 3.3). Finally, we introduce a two-stage training paradigm: in the first stage, Diversity-Preserving Policy Optimization (DiPO) prevents entropy collapse and maintains trajectory diversity during reinforcement learning, and in the second stage, DAPO anneals the policy toward exploitation to consolidate performance (Section 3.4).

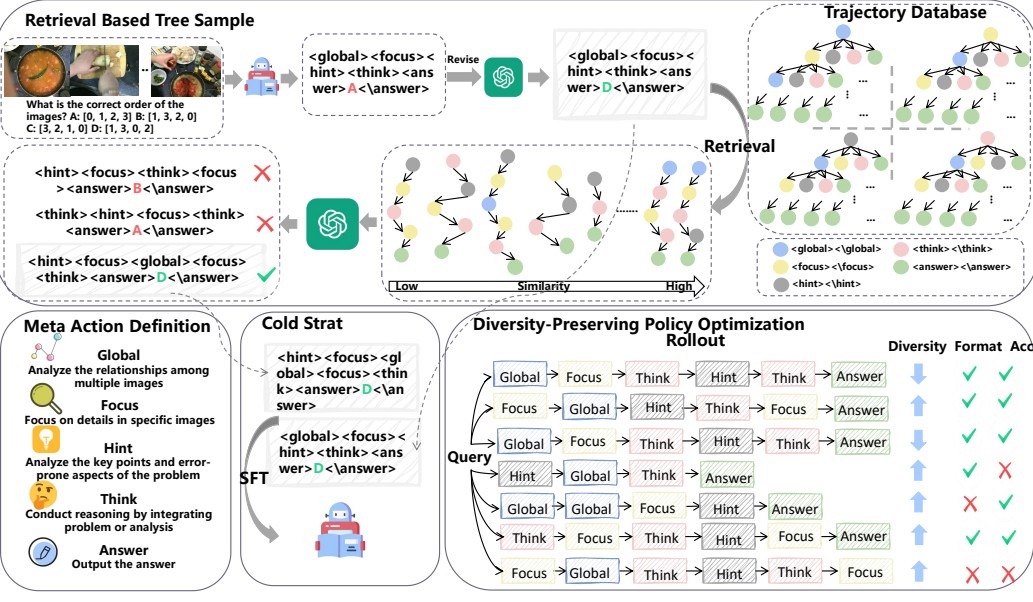

Figure 1: Overview of CINEMA.

## 3.1 META ACTION DEFINITION

**Global.** This meta action simulates how humans typically approach complex problems by first reading through the entire question to grasp its overall structure. When dealing with multi-image input tasks, there may be temporal, spatial, semantic, or other relationships between the images. This action helps the model identify and leverage these inter-image dependencies to enhance understanding and reasoning.

**Focus.** This meta action simulates how humans tackle complex problems by concentrating on analyzing key information relevant to the question. In the context of multi-image reasoning, critical clues may reside in a specific image. The model should therefore focus its analysis on that image and pay close attention to salient visual details.

**Hint.** This meta action simulates how humans improve accuracy by summarizing key points and error-prone aspects of a problem when solving tasks. In multi-image reasoning tasks, similarly, there often exists misleading or easily confusable information between images.

**Think.** This meta action simulates how humans engage in internal reasoning by actively processing acquired information to formulate solutions or hypotheses. It involves analyzing the relationships between provided clues, leveraging prior knowledge, and performing logical inference.

**Answer.** This meta action outputs the final answer based on all prior analytical insights and reasoning outcomes. It is the final action in the compliant trajectory.

## 3.2 RETRIEVAL BASED TREE SAMPLE

To effectively leverage the defined meta actions for multi-image reasoning enhancement, we propose a novel cold-start data sampling strategy called Retrieval-Based Tree Sampling. This approach is inspired by human learning mechanisms, where students first attempt problems independently before receiving guidance from teachers who first refine their initial approach and then introduce alternative solution pathways.

With the meta actions defined in Section 3.1, we maintain several meta action trees, each containing multiple reasoning trajectories. Every trajectory in these trees terminates with the "Answer" meta action, forming complete reasoning paths from problem comprehension to solution derivation. These trees serve as a database of diverse reasoning strategies that can be retrieved and adapted for new problems. The Retrieval-Based Tree Sampling process is shown as follows:

**Step 1. Initial Trajectory Generation.** We first prompt a smaller model (student model) to perform initial reasoning on the given task using meta actions. This generates an initial trajectory regardless of whether the final answer is correct or incorrect. This step mirrors how students first attempt to solve problems using their existing knowledge and reasoning patterns.

**Step 2. Teacher-Guided Trajectory Refinement.** The initial trajectory from Step 1 is then provided to a stronger model (GPT-4o, serving as the teacher model). The teacher model follows the student's reasoning action trajectories and reason again, similar to how human teachers guide students by first understanding their thought processes and then providing corrections. This produces a correct trajectory that maintains the original action trajectories while ensuring accuracy.

**Step 3. Retrieval-Based Diverse Sampling.** To enrich the learning experience and expand the exploration space for subsequent reinforcement learning, we perform retrieval-based sampling from our trajectory tree database. Starting from trajectories with low similarity to the initial trajectory from Step 2, we progressively search through increasingly similar trajectories until we identify an alternative correct reasoning path. This process ensures that each training instance is associated with two distinct correct trajectories.

## 3.3 DATASET CONSTRUCTION

To train our model, we construct a high-quality training dataset that supports both cold-start initialization and reinforcement learning phases. Our dataset encompasses three primary categories: multi-image tasks in which the number of input images is at least two, multi-frame tasks that involve reasoning over sequential frames from videos or time-series visual data, and single image

tasks that in which the number of input image is only one. All the data is obtained through existing open-source multi-modal datasets. More details about dataset is shown in Appendix A.5.

The key distinction between our cold-start and reinforcement learning dataset splits lies in the trajectory generation process described in Section 3.2. Cold-start training data consists of problems where GPT-4o successfully provides correct answers during Step 2, and for these instances, we proceed to Step 3 (retrieval-based diverse sampling) to obtain two distinct correct reasoning trajectories per problem that serve as supervised learning targets for cold start training. In contrast, reinforcement learning data comprises problems where GPT-4o fails to produce correct answers during Step 2, and these challenging cases are reserved for reinforcement learning.

### 3.4 Balancing Exploration and Exploitation via Two-Stage Optimization

A critical challenge in reinforcement learning for reasoning is policy entropy collapse which limits exploration and generalization capacity. We address this through a two-stage training paradigm: first maintaining trajectory diversity to preserve exploration, then gradually shifting toward exploitation to consolidate performance.

**Diversity-Preserving Policy Optimization (DiPO).** In the first stage, we aim to prevent entropy collapse by maintaining diversity at the meta-action level. To this end, we propose DiPO which is build on DAPO (Yu et al., 2025)(more details about DAPO is shown in Appendix A.2). Our central hypothesis is that encouraging a variety of solution strategies can better leverage the model's potential and improve its generalization performance like human.

We operationalize this by promoting diverse responses for questions that the model answers correctly. To this end, we define the reward as a weighted combination of accuracy and format validity:

$$R = 0.5 \cdot \left[ R_{\text{acc}} \cdot \left( R_{\text{acc}} - \frac{N-1}{G-1} \cdot 0.1 \right) \right] + 0.5 \cdot R_{\text{format}}, \tag{1}$$

where $R_{\text{acc}}$ and $R_{\text{format}}$ are binary indicators:

$$R_{\text{acc}} = \begin{cases} 1, & \text{if the answer is correct,} \\ 0, & \text{otherwise,} \end{cases} \qquad R_{\text{format}} = \begin{cases} 1, & \text{if all meta actions in the response are valid,} \\ 0, & \text{otherwise.} \end{cases}$$

Here, $G$ denotes the group size used in sampling, and $N$ represents the number of trajectories within the group that share identical meta-action patterns. Intuitively, the penalty term $\frac{N-1}{G-1}$ discourages over-reliance on homogeneous trajectories, thereby encouraging the model to maintain diversity across solutions. This design ensures that correct answers are not only accurate but also exhibit a broad spectrum of solution strategies, thereby enhancing the model's generalization. In practice, to perform dynamic sampling as in DAPO, we use $R_{\text{acc}}$ as the filtering criterion rather than the combined reward $R$.

**Annealed Exploitation.** In the second stage, we employ DAPO with an annealing schedule to gradually shift from exploration to exploitation, leveraging the diversity obtained in stage one while consolidating performance gains. This two-stage approach maintains higher entropy levels throughout training compared to standard methods, as validated by our Pass@K experiments.

## 4 Experiment Setup

### 4.1 Benchmark and Baselines

**Benchmark.** To ensure a comprehensive evaluation, we examine the performance of our method across a broad spectrum of benchmarks, encompassing both multi-image and single-image types. Specifically, for multi-image evaluations, we cover **multi-image comprehensive benchmarks** (including MUIR (Wang et al., 2025a), MMIU (Meng et al., 2025b), and Mantis-Eval (Jiang et al.)), **multi-image reasoning benchmarks** (including MV-MATH (Wang et al., 2025b), MIRB (Zhao et al., 2024) and EMMA (Hao et al.)), **video comprehensive benchmarks** (including MVBench (Li et al., 2024c), and VideoMME (Fu et al., 2025)) and **video reasoning benchmarks** (including

| Model | MUIR | MMIU | MVMATH | EMMA | MIRB | Mantis | MVBench | VideoMME | VideoMMMU | Overall |
|---|---|---|---|---|---|---|---|---|---|---|
| *Closed-Source MLLMs* | | | | | | | | | | |
| GPT-4V | - | - | 24.5 | - | - | 62.7 | 43.5 | 59.9 | - | - |
| Gemini-1.5-Pro | - | - | 29.1 | - | - | - | - | 71.9 | 53.9 | - |
| GPT-4o | 68.0 | 55.7 | 32.1 | 32.7 | - | - | - | 75.0 | 61.2 | - |
| *Open-Source General MLLMs* | | | | | | | | | | |
| OpenFlamingo-v2 9B | 22.3 | 23.7 | - | - | 28.8 | 12.4 | 7.9 | - | - | - |
| LLaVA 1.6 7B | 27.4 | 22.2 | - | - | 29.8 | 45.6 | 40.9 | - | - | - |
| VILA1.5 8B | 33.1 | 32.5 | - | - | 36.5 | 51.2 | 49.4 | 20.9 | - | - |
| LLaVA-OneVision 7B | 41.8 | 40.3 | 19.1 | - | 51.2 | 64.2 | 56.7 | - | - | - |
| InternVL2.5 8B | 51.1 | 46.7 | 18.8 | 21.0 | 52.5 | 67.7 | **72.0** | 56.1 | 35.2 | 46.8 |
| Qwen2.5-VL-7B | 57.9 | 50.6 | 26.7 | 20.4 | 48.3 | 64.5 | 62.6 | 56.7 | 45.8 | 48.2 |
| *Multi-image/Video Enhancing MLLMs* | | | | | | | | | | |
| Mantis-Idefics2 8B | 44.5 | 45.6 | 15.5 | 20.3 | 34.8 | 57.1 | 51.4 | 42.6 | 19.3 | 36.8 |
| LLaVA-NeXT-Interleave 7B | 31.1 | 47.3 | 14.7 | 19.0 | 39.3 | 62.7 | 53.1 | 47.2 | 23.2 | 37.5 |
| mPLUG-Owl3 8B | 34.0 | 39.7 | 18.7 | 24.8 | 41.2 | 63.1 | 54.5 | 53.5 | 32.0 | 40.2 |
| MIA-DPO 7B | - | - | - | - | - | 60.4 | 63.6 | - | - | - |
| CcDPO 7B | 44.8 | - | - | - | **60.7** | 69.1 | - | - | - | - |
| VISC 7B | 44.5 | 52.8 | - | - | 60.2 | 69.1 | 68.0 | - | - | - |
| VideoR1 7B | - | - | - | - | - | - | 63.6 | 57.4 | 49.8 | - |
| VideoRFT 7B | 56.6 | 44.5 | 25.1 | 17.8 | 46.7 | 56.7 | 62.1 | 59.8 | 51.1 | 46.7 |
| TW-GRPO 7B | 55.9 | 44.9 | 28.2 | 22.5 | 24.3 | 49.8 | 63.3 | 55.1 | 40.8 | 42.8 |
| Ours | **71.6** | **53.3** | **36.9** | **29.3** | 55.2 | 67.7 | 66.5 | 59.4 | 49.0 | **54.3** |
| $\Delta$ (vs Qwen2.5VL 7B) | +13.7 | +2.7 | +10.2 | +8.9 | +6.9 | +3.2 | +3.9 | +2.7 | +3.2 | +6.1 |
| Ours [with DiPO] | 67.9 | 52.2 | 35.1 | 28.4 | 54.4 | **71.0** | 67.1 | 60.2 | **51.6** | 54.2 |
| $\Delta$ (vs Qwen2.5VL 7B) | +10.0 | +1.6 | +8.4 | +8.0 | +6.1 | +6.5 | +4.5 | +3.5 | +5.8 | +6.0 |
| Ours [with DiPO and annealing] | 71.0 | 52.2 | 35.0 | 28.6 | 55.7 | 68.4 | 66.8 | **61.0** | 50.1 | **54.3** |
| $\Delta$ (vs Qwen2.5VL 7B) | +13.1 | +1.6 | +8.3 | +8.2 | +7.4 | +3.9 | +4.2 | +4.3 | +4.3 | +6.1 |

Table 1: Performance on multi-image/video benchmark. Ours indicates training with DAPO, Ours [with DiPO] indicates training with DiPO, and Ours [with DiPO reward and annealing] indicates training with two-stage RL, where all models are trained for the same number of steps.

VideoMMMU (Hu et al., 2025b)). For single-image evaluations, we include **single-image comprehensive benchmarks** (including MMMU-Pro (Yue et al., 2024) and M3COT (Chen et al., 2024a)) as well as **mathematics reasoning benchmarks** (including MM-Math (Sun et al., 2024), Math-Vision (Wang et al., 2024b), and MathVista (Lu et al., 2023)). Accuracy is reported as the evaluation metric for all these benchmarks.

**Baslines.** We compare our method's performance against four categories of models: closed-source MLLMs, including GPT-4V (Achiam et al., 2023), Gemini-1.5-Pro (Team et al., 2023), and GPT-4o (Hurst et al., 2024); open-source general-purpose MLLMs, including OpenFlamingo-v2 (Awadalla et al., 2023), LLaVAv1.6 (Liu et al., 2024), VILA1.5 (Lin et al., 2024), LLaVA-OneVision (Li et al.), InternVL2.5 (Chen et al., 2024b), and Qwen2.5-VL (Bai et al., 2025); and multi-image/video enhanced models, including Mantis-Idefics (Jiang et al.), mPLUG-Owl3 (Ye et al., 2025), LLaVA-NeXT-Interleave (Li et al., 2024a), CMMCOT (Zhang et al., 2025a), MIA-DPO (Liu et al., 2025c), VISC (Zhang et al., 2025b), VideoR1 (Feng et al., 2025), and VideoRFT (Wang et al., 2025c); Single-image reasoning models: Mulberry 7B (Yao et al., 2024), R1-Onevision 7B Yang et al. (2025b), VLAA-Thinker 7B (Chen et al., 2025a), VisonR1 7B (Huang et al., 2025), MixedR1 7B (Xu et al., 2025).

## 4.2 IMPLEMENTATION DETAILS

We select Qwen2.5VL 7B as our backbone model. During the cold-start training phase, the model is initialized and trained for two epochs with a learning rate of $1 \times 10^{-5}$. We employ a two-stage reinforcement learning procedure. The first stage consists of 700 steps of DiPO-based entropy enhancement, followed by 300 steps of DAPO-based annealed exploitation. In the subsequent reinforcement learning stage, both the KL-divergence and entropy regularization terms are omitted. Rollouts are generated using a batch size of 64, a temperature of 1.0, and 8 rollouts per prompt. For policy optimization, an update batch size of 32 is adopted. Regarding reward design, we incorporate domain-specific validation mechanisms: math_verify (Kydlíček) and mathruler (hiyouga, 2025) are employed to evaluate answers in mathematical problem-solving, whereas exact string matching is applied to non-mathematical tasks. To ensure structural consistency, format rewards are introduced by imposing constraints on the response space, requiring outputs to conform to a valid meta-action trajectory. Specifically, for single-image inputs, the global action is disallowed, whereas for multi-image inputs, the inclusion of the global action is mandatory. During inference,

we set the decoding hyperparameters as follows: temperature $= 0.6$, top-$p = 0.7$, and a maximum of 1024 generated tokens. Additional implementation details are provided in the Appendix A.4.

## 5 EXPERIMENTS

### 5.1 RESULTS ON MULTI-IMAGE BENCHMARK

Table 4.1 presents the experimental results on multi-image benchmarks, where our model demonstrates significant improvements over Qwen2.5VL across all benchmarks, achieving state-of-the-art performance on MUIR, MVMath, EMMA, VideoMME, and VideoMMMU benchmarks. Remarkably, our model surpasses the closed-source GPT-4o on both MUIR and MVMath benchmarks. On multi-image comprehensive benchmarks, our model achieves 13.7% improvement over Qwen2.5VL on the MUIR benchmark and 6.9% improvement on MIRB. These multi-image benchmarks encompass diverse multi-image tasks, demonstrating our model's robust capability in processing multi-image inputs. On multi-image resoning benchmarks, MVMath is a mathematics dataset with multi-image inputs, while EMMA encompasses multiple academic disciplines. These benchmarks require strong reasoning capabilities from the model. Our model achieves 10.2% improvement on MVMath and 8.9% improvement on EMMA, reflecting enhanced reasoning capabilities attributed to CINEMA, which simulates human-like reasoning processes through structured meta-action trajectory and cross-image relationship modeling. Notably, our model surpasses specialized video reasoning models across all three video benchmarks, despite not being specifically designed for video reasoning tasks. This demonstrates our model's superior performance in handling temporal multi-image data, suggesting that our approach effectively captures both spatial and temporal dependencies inherent in sequential visual information.

### 5.2 RESULTS ON SINGLE-IMAGE BENCHMARK

Table 2 presents the results on single-image benchmarks, where our model demonstrates equally strong capabilities. Our model achieves superior overall performance compared to existing models specifically designed for single-image reasoning, despite being trained on only a limited amount of single-image data. This validates the generalizability of our approach, proving its effectiveness not only for multi-image scenarios but also for single-image tasks. On comprehensive single-image benchmarks, our model achieves 3% improvement on MMMU-Pro and 3.8% improvement on M3COT, surpassing the closed-source models GPT-4V and GPT-4o. On mathematical benchmarks, our model attains state-of-the-art performance on MM-Math and achieves comparable performance to existing models specialized in single-image reasoning across other mathematical benchmarks. In single-image scenarios, the model trained with the two-stage RL approach outperforms standard DAPO and DiPO, indicating that the two-stage training achieves a better exploration-exploitation trade-off, thereby promoting improved generalization across diverse tasks.

| Model | MMMU-Pro | M3COT | MM-IQ | MM-Math | Math-Vision | MathVista | MathVerse | Overall |
|---|---|---|---|---|---|---|---|---|
| *Closed-Source MLLMs* | | | | | | | | |
| GPT-4V | - | 62.60 | - | 23.1 | 22.76 | 49.9 | 39.4 | - |
| Gemini-1.5-Pro | 51.47 | - | 26.86 | - | - | - | - | - |
| GPT-4o | 56.13 | 55.7 | 26.87 | 31.8 | - | - | - | - |
| *Open-Source General MLLMs* | | | | | | | | |
| LLaVA-OneVision 7B | - | - | - | - | - | 63.2 | 26.2 | - |
| InternVL2.5 8B | 34.3 | - | - | - | 22.0 | 64.4 | 39.5 | - |
| Qwen2.5VL 7B | 38.0 | 60.1 | 26.1 | 36.4 | 19.5 | 65.3 | 40.4 | 40.8 |
| *Reasoning MLLMs* | | | | | | | | |
| Mulberry | - | - | - | 23.7 | - | 63.1 | - | - |
| R1-Onevision 7B | 33.9 | 57.3 | 25.1 | 32.9 | 29.9 | 64.1 | 46.4 | 41.4 |
| VLAA-Thinker 7B | 39.5 | 61.3 | 26.3 | 39.0 | 26.4 | 68.0 | 47.8 | 44.0 |
| VisonR1 7B | 30.3 | 53.2 | 24.3 | 40.0 | 29.9 | **73.5** | **52.4** | 43.4 |
| MixedR1 7B | 38.0 | 59.9 | 25.9 | 35.8 | **30.3** | 70.6 | 40.8 | 43.0 |
| Ours | 40.6 | 63.5 | 25.6 | **43.8** | 26.7 | 68.7 | 49.4 | 45.5 |
| Δ (vs Qwen2.5VL 7B) | +2.6 | +3.4 | +0.5 | +7.4 | +7.2 | +3.4 | +9.0 | +4.7 |
| Ours [with DiPO] | 40.7 | **63.9** | 26.3 | 43.4 | 26.1 | 70.0 | 47.6 | 45.4 |
| Δ (vs Qwen2.5VL 7B) | +2.7 | +3.8 | +0.2 | +7.0 | +6.6 | +4.7 | +7.2 | +4.6 |
| Ours [with DiPO reward and annealing] | **41.0** | 62.7 | **27.3** | 43.4 | 26.1 | 70.1 | 48.5 | **45.6** |
| Δ (vs Qwen2.5VL 7B) | +3.0 | +2.6 | +1.2 | +7.0 | +4.8 | +8.1 | +8.1 | +4.8 |

Table 2: Performance on single-image benchmark.

## 5.3 RESULTS ON PASS@K SETTING

To further validate the advantages of our proposed two-stage RL approach, we conduct Pass@K experiments on 7 multi-image and 7 single-image benchmarks, comparing models with and without DiPO and annealing. We evaluate the accuracy across K inference attempts, where $K \in 2, 4, 8, 16$, and a model is considered correct if at least one inference attempt produces the correct answer. We report the average accuracy in Figure 2. The results show that incorporating DiPO and annealing consistently outperforms the baseline across pass@2, pass@4, pass@8, and pass@16, further demonstrating the effectiveness of our two-stage RL method. After this training paradigm, the model exhibits more diverse sampling behavior and achieves a higher performance ceiling.

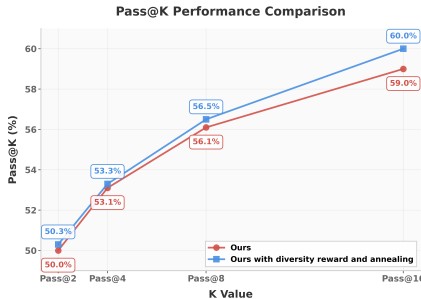

Figure 2: Pass@K performance.

| Method | MUIR | | MMIU | | EMMA | |
|---|---|---|---|---|---|---|
| Original | 57.9 | | 50.6 | | 20.4 | |
| Direct Prompting | 33.8 | | 36.9 | | 14.1 | |
| | SFT | RL | SFT | RL | SFT | RL |
| Conventional CoT | **59.0** | 70.0 | 49.9 | 51.6 | 21.2 | 26.9 |
| Single Trajectory | 56.3 | 65.1 | 50.9 | 52.2 | 24.0 | 27.9 |
| **Ours (Two Traj.)** | 58.2 | **71.6** | **51.9** | **53.3** | **24.8** | **29.3** |

Table 3: Ablation study on Retrieval-Based Tree Sampling strategy.

## 5.4 FUTHER ANALYSIS

To conduct an in-depth analysis of our model's effectiveness, we propose 4 research questions and conduct detailed experiments:

**RQ1:** Can diverse trajectories improve model performance?

**RQ2:** How does the model perform with different numbers of input images?

**RQ3:** How does the model perform across different tasks?

**RQ4:** How does two-stage RL training influence entropy control and training dynamics?

**About RQ1: Can diverse trajectories improve model performance?**

To validate the effectiveness of our proposed Retrieval-Based Tree Sampling strategy, which samples two different trajectories for each data point, we conduct comparative experiments on three benchmarks: MUIR, MMMU and EMMA. We compare against three baselines: (1) cold start training with only one trajectory then RL; (2) cold start training with conventional Chain-of-Thought (CoT) in the format of: `<think>reasoning here</think><answer>answer here</answer>` then RL; and (3) directly prompting MLLMs to perform reasoning using meta actions without additional training. The experimental results are shown in Table 3. The model trained with two trajectories achieves superior average performance compared to models trained with single trajectories and conventional CoT training. Moreover, the best results under RL are all achieved by the model trained with two trajectories. In comparison to the directly prompted model, we observe that the untrained model performs poorly in utilizing our defined meta actions, showing significant performance degradation relative to the original model. This demonstrates the necessity of constructing datasets for subsequent training.

**About RQ2: How does the model perform with different numbers of input images?**

To investigate our model's capability in processing varying numbers of images, we conduct experiments on MUIR and MMIU benchmarks. The MUIR dataset contains samples with 2-9 input images per instance, whereas MMMU contains samples with 2-32 input images per instance. The experimental results are presented in Figures 3a and 3b. For different numbers of input images, our model outperforms the base model in most cases. Even when the number of input images exceeds 17, our model still achieves a significant improvement. This demonstrates the strong capability of our model in handling multi-image inputs and validates the effectiveness of the proposed cognition-inspired reasoning framework.

**About RQ3: How does the model perform across different tasks?**

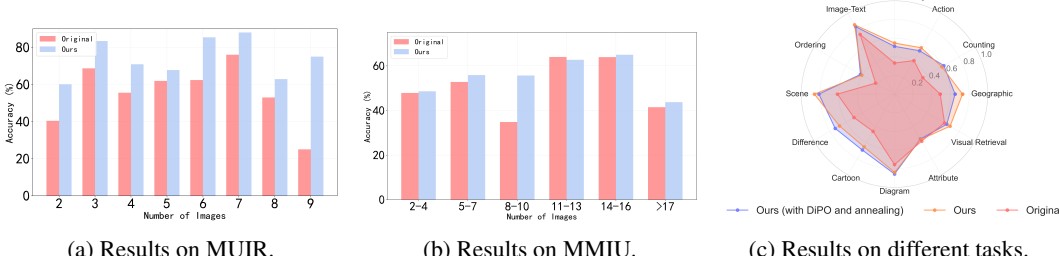

(a) Results on MUIR.      (b) Results on MMIU.      (c) Results on different tasks.

Figure 3: Results about RQ2 and RQ3.

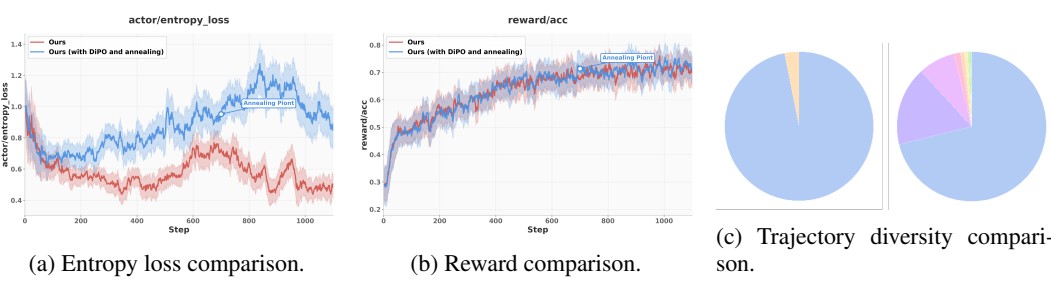

(a) Entropy loss comparison.      (b) Reward comparison.      (c) Trajectory diversity comparison.

Figure 4: Results about RQ4.

To explore the performance of our model across different tasks, we present the results on MUIR, which consists of 12 distinct tasks, in Figure 3c. Our model achieves improvements on almost all tasks. Notably, tasks such as Geographic, Cartoon, and Visual Grounding were not included in our training set, yet our model still yields significant improvements on these tasks. This further demonstrates the generalization ability of our proposed reasoning framework in multi-image tasks.

**About RQ4: How does two-stage RL training influence entropy and training dynamics?**

To explore how two-stage RL training influences entropy control and training dynamics, we present the results in Figure 4a and 4b. In the first stage, DiPO maintains a moderate entropy level, which gradually decreases in the second annealing stage. Compared to the baseline, DiPO consistently preserves higher entropy, effectively preventing entropy collapse. This sustained entropy encourages exploration, avoids over-concentration, and retains diversity in meta-actions, thereby reducing the risk of premature convergence. Importantly, despite maintaining higher entropy, DiPO achieves comparable training accuracy to the baseline, demonstrating that the entropy-preserving mechanism does not harm training performance. In Figure 4c, each color represents one type of trajectory. The left one is the visualization without DiPO and annealing, and the right one is with DiPO and annealing. We show that even after the annealing stage, our model continues to promote richer and more diverse meta-actions during generation, thereby sustaining exploration and preserving policy diversity throughout training. The effectiveness of this two-stage training is further supported by the Pass@K results in Figure 2.

## 6 CONCLUSION

In this work, we introduce CINEMA, a cognition-inspired meta-action framework that systematically decomposes multi-image reasoning into structured cognitive steps. By leveraging Retrieval-Based Tree Sampling for cold-start training and a two-stage reinforcement learning paradigm with DiPO and annealed DAPO, our approach effectively effectively improve multi-image reasoning ability. Extensive experiments across multi-image, video, and single-image benchmarks demonstrate that CINEMA not only achieves state-of-the-art performance, surpassing even large general-purpose models such as GPT-4o in some key benchmarks, but also maintains higher policy diversity and adaptability. These results highlight the effectiveness, scalability, and generalizability of our framework, paving the way toward more robust multimodal reasoning systems.

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

# A APPENDIX

## A.1 AI WRITING ASSISTANCE STATEMENT

Large language models (e.g., ChatGPT) were used exclusively for minor language refinement, such as improving phrasing and readability. They were not involved in generating scientific content, conducting experiments, or performing analyses. The authors are entirely responsible for all ideas, results, and conclusions presented in this paper.

## A.2 BACKGROUND: DAPO

DAPO is an improved variant of GRPO, which directly computes the advantage $A_t$ using the average reward over multiple sampled outputs, thereby eliminating the need for a separate value function as in PPO. Specifically, given a prompt $\mathbf{q} \sim P(Q)$, we sample $G$ rollouts $\{\mathbf{o}_i\}_{i=1}^G$ from the current policy $\pi_{\theta_{\text{old}}}$. At each token position $t$ in rollout $i$, the likelihood ratio is defined in Eq. 2.

$$r_{i,t}(\theta) = \frac{\pi_\theta\left(o_{i,t} \mid \mathbf{q}, \mathbf{o}_{i,<t}\right)}{\pi_{\theta_{\text{old}}}\left(o_{i,t} \mid \mathbf{q}, \mathbf{o}_{i,<t}\right)} \tag{2}$$

The group-relative advantage $\hat{A}_{i,t}$ is then obtained by standardizing each return $R_i$ within the group, defined in Eq. 3.

$$\hat{A}_{i,t} = \frac{R_i - \text{Mean}\left(\{R_j\}_{j=1}^G\right)}{\text{Std}\left(\{R_j\}_{j=1}^G\right)}. \tag{3}$$

In contrast to GRPO, DAPO introduces several methodological advancements. Specifically, it employs a Clip-Higher mechanism, wherein $\epsilon_{\text{high}}$ is set greater than $\epsilon_{\text{low}}$ to enhance exploratory behavior; integrates Dynamic Sampling to systematically exclude data instances lacking informative learning signals; incorporates an Overlong Punishment strategy to constrain excessively verbose outputs; and adopts a Token-level Loss formulation to mitigate the inherent bias between responses of varying lengths. The training then proceeds by maximizing the clipped surrogate objective, defined for DAPO as follows:

$$\mathcal{J}_{\text{DAPO}}(\theta) = \mathbb{E}_{(q,a)\sim\mathcal{D}, \{o_i\}_{i=1}^G \sim \pi_{\theta_{\text{old}}}(\cdot|q)}$$
$$\left[\frac{1}{\sum_{i=1}^G |o_i|} \sum_{i=1}^G \sum_{t=1}^{|o_i|} \min\left(r_{i,t}(\theta)\hat{A}_{i,t}, \text{clip}\left(r_{i,t}(\theta), 1-\varepsilon_{\text{low}}, 1+\varepsilon_{\text{high}}\right)\hat{A}_{i,t}\right)\right], \tag{4}$$
$$\text{s.t.} \quad 0 < \left|\left\{o_i \mid \text{is\_equivalent}(R_i, 1)\right\}\right| < G.$$

## A.3 BENCHMARK

This section provides a detailed description of the benchmark used for evaluation.

**MUIR** MUIRBENCH (Wang et al., 2025a) is a comprehensive benchmark designed for robustly evaluating MLLMs' multi-image understanding capabilities. It comprises 11,264 images and 2,600 multiple-choice questions (average 4.3 images per instance), covering 12 diverse multi-image tasks (e.g., action understanding, cartoon storytelling, geographic map reasoning, 3D object multiview retrieval).

**MMIU** The Multimodal Multi-image Understanding (MMIU) (Meng et al., 2025b) is a comprehensive benchmark tailored for evaluating MLLMs on multi-image comprehension tasks. Structured around cognitive psychology, it enumerates 7 types of multi-image relationships (refined from semantic, temporal, spatial categories) and covers 52 diverse tasks (e.g., multi-view action recognition, 3D object detection) . In terms of scale, MMIU includes 77,659 images (2–32 per instance, averaging 6.64) and 11,698 meticulously curated multiple-choice questions.

**MV-MATH** MV-MATH (Wang et al., 2025b) is a specialized benchmark designed to evaluate MLLMs on mathematical reasoning in multi-visual contexts—addressing the gap in existing benchmarks that mostly focus on single images. It comprises 2,009 high-quality mathematical problems derived from real K-12 scenarios.

**EMMA** EMMA (Hao et al.) is a benchmark designed to evaluate Multimodal LLMs on genuine cross-modal reasoning. Its 2,788 questions across math, physics, chemistry, and coding require integrated visual-textual understanding, preventing solutions based on shallow cues or text alone.

**Mantis-Eval**  Mantis-Eval (Jiang et al.) is a benchmark dataset designed to evaluate a model's ability to reason across multiple images. It contains 217 challenging examples.

**MIRB**  MIRB (Zhao et al., 2024) is a dedicated dataset addressing the gap in evaluating vision-language models (VLMs) on multi-image understanding, as existing benchmarks focus primarily on single-image inputs. It encompasses 925 samples across four core dimensions: perception, visual world knowledge, reasoning, and multi-hop reasoning, with all tasks requiring cross-comparison of multiple images (ranging from 2 to 42, averaging 3.78 per question).

**MVBench**  MVBench (Li et al., 2024c) is a multi-modal video benchmark addressing the lack of temporal understanding evaluation in MLLMs, covering 20 multi-frame-dependent video tasks (defined via a static-to-dynamic method). It is built efficiently by auto-converting public video annotations into multiple-choice QA (with ground-truth for fairness), reveals existing MLLMs' poor temporal understanding.

**Video-MME**  Video-MME (Fu et al., 2025) is the first comprehensive benchmark designed to evaluate Multi-modal Large Language Models (MLLMs) in video analysis. It fills the gap in assessing the understanding of sequential visual data by featuring 900 videos (ranging from 11 seconds to 1 hour) across 6 core domains (e.g., Knowledge, Sports Competition) and 30 subfields. Each video is paired with three expert-annotated multiple-choice QA pairs, resulting in a total of 2,700 questions. To support multi-modal reasoning, the benchmark also provides subtitles for 744 videos and audio tracks for all 900 videos.

**Video-MMMU**  Video-MMMU (Hu et al., 2025b) is a benchmark designed to evaluate the knowledge acquisition capabilities of large multimodal models (LMMs) from professional video content. It comprises 300 expert-level videos (average length 506.2 seconds) spanning six disciplines (e.g., Art, Business) and 30 subfields, paired with 900 human-annotated question–answer pairs (three per video). The benchmark measures performance across three cognitive stages: (1) *Perception*, assessing whether models can extract salient knowledge-related details from video content; (2) *Comprehension*, evaluating the ability to grasp and reason about the underlying concepts; and (3) *Adaptation*, examining whether models can transfer the acquired knowledge to novel or unfamiliar scenarios.

**MMMU-Pro**  MMMU-Pro (Yue et al., 2024) is an enhanced version of the MMMU benchmark, designed to more rigorously evaluate multimodal models' understanding and reasoning. It filters out text-only solvable questions, augments candidate options, and embeds questions within images, forcing models to both "see" and "read." Results show substantially lower performance (16.8%–26.9%), highlighting its difficulty and realism, and providing a more robust evaluation framework for future multimodal reasoning research.

**M3CoT**  M3CoT (Chen et al., 2024a) addresses gaps in existing MCoT benchmarks (lack of visual reliance, single-step reasoning, limited domains) by enabling multi-domain, multi-step, multi-modal reasoning across 3 domains (science, mathematics, commonsense), 17 topics, and 263 categories. It has 11,459 total samples (7,973 train, 1,127 dev, 2,359 test) with diverse image types (geographic graphs, health images, etc.).

**MM-MATH**  MM-MATH(Sun et al., 2024)consists of 5,929 open-ended middle school math problems paired with visual contexts, and it adopts fine-grained classification covering three dimensions: difficulty, grade level, and knowledge points. Unlike existing benchmarks that depend solely on binary answer comparison, MM-MATH incorporates both outcome evaluation and process evaluation. Specifically, the process evaluation utilizes an LMM-as-a-judge to automatically analyze the steps of solutions, as well as identify and categorize errors into specific types.

**MathVista**  MathVista (Lu et al., 2023) is proposed as a benchmark integrating challenges from mathematical and visual tasks. It contains 6,141 examples, sourced from 28 existing multimodal math datasets and 3 new ones (IQTest, FunctionQA, PaperQA), requiring fine-grained visual understanding and compositional reasoning—tasks that state-of-the-art foundation models find challenging.

**MATH-V**  MATH-V (Wang et al., 2024b) is a curated dataset designed to address the limited question diversity and subject breadth of existing visual math reasoning benchmarks (e.g., MathVista).

It comprises 3,040 high-quality math problems with visual contexts, all sourced from real math competitions. The dataset spans 16 distinct mathematical disciplines and includes 5 graded difficulty levels, offering comprehensive, diverse challenges for evaluating Large Multimodal Models (LMMs)' mathematical reasoning abilities. Additionally, MATH-V reveals a notable performance gap between current LMMs and humans, while its detailed categorization supports thorough error analysis of LMMs to inform future research.

### A.4 TRAINING DATA CONSTRUCTION

For the construction of the training dataset, we referenced Mantis (Jiang et al.), LLaVA-Interleave (Li et al., 2024a), Leopard Jia et al., and VideoR1 Feng et al. (2025). Overall, our dataset consists of multi-image data and single-image data, with 57k samples for cold-start training and 58k samples for RL. The detailed dataset statistics are presented in Table 1. Regarding the partitioning criteria for RL data and cold-start data, the key distinction between our cold-start and reinforcement learning dataset splits lies in the trajectory generation process described in Section 3.2. Cold-start training data consists of problems where GPT-4o successfully provides correct answers during Step 2, and for these instances, we proceed to Step 3 (retrieval-based diverse sampling) to obtain two distinct correct reasoning trajectories per problem that serve as supervised learning targets for cold start training. In contrast, reinforcement learning data comprises problems where GPT-4o fails to produce correct answers during Step 2, and these challenging cases are reserved for reinforcement learning.

| Type | Dataset | Count for SFT | Count for RL |
|---|---|---|---|
| Multi-Image | ChartVQA(Jia et al.) | 2501 | - |
| | SlideVQA(Jia et al.) | 3249 | 3000 |
| | ALFRED(Shridhar et al., 2020) | 8357 | 2754 |
| | Nuscenes(Bansal et al., 2020) | 580 | 4946 |
| | RecipeQA(Yagcioglu et al., 2018) | 8759 | 5069 |
| | IconQA(Lu et al., 2021b) | 5315 | 3000 |
| | nlvr2(Suhr et al., 2018) | 5424 | 1620 |
| | Spot-the-Diff(Jhamtani & Berg-Kirkpatrick, 2018) | 2248 | 2589 |
| | LRV(Liu et al., 2023) | - | 2993 |
| | RAVEN(Zhang et al., 2019) | - | 3200 |
| Video | Star(Wu et al., 2024) | 5490 | 2754 |
| | NextQA(Xiao et al., 2021) | 1193 | 3000 |
| | Clevrer(Yi et al., 2019) | 3047 | 4478 |
| | Perception(Patraucean et al., 2023) | 2964 | 2500 |
| Single-Image | Clevr_cogen_a_train[1] | 1506 | - |
| | Clevr_CoGenT_TrainA_70K_Complex[2] | 1159 | 3000 |
| | M3COT(Chen et al., 2024a) | 1147 | - |
| | Share-GRPO(Yao et al., 2025) | 1145 | 3000 |
| | GEOQA_R1V_Train_8K[3] | 800 | 4816 |
| | AI2D(Kembhavi et al., 2016) | 630 | - |
| | MMK12(Meng et al., 2025a) | 442 | 3537 |
| | Geometry3k(Lu et al., 2021a) | 317 | 1406 |
| | ScienceQA(Lu et al., 2022) | 259 | - |
| | PISC(Li et al., 2017) | 244 | - |
| | Geoqa+(Cao & Xiao, 2022) | 172 | 891 |
| | GQA(Hudson & Manning, 2019) | 119 | - |
| | CLEVR_v1.0(Johnson et al., 2017) | 118 | - |
| | COCO(Lin et al., 2014) | 78 | - |
| | LRV(Liu et al., 2023) | - | 3063 |

Table 4: Statistics of Training Data

### A.5 IMPLEMENTATION

Our SFT experiments are primarily conducted using the LLaMA Factory framework (Zheng et al., 2024), with the main hyperparameters summarized in Table 5. For the RL stage, we rely on the EasyR1 framework (Yaowei Zheng, 2025), a multi-model large-scale training system built upon VERL (Sheng et al., 2024), and the key parameters are reported in Table 6.

### A.6 CASE STUDY

Here we present a case study of our model in Figure 5 and 6, covering multi-image benchmarks, video benchmarks, and single-image benchmarks. The results demonstrate that, across different types of tasks, our model can dynamically invoke appropriate meta-actions to analyze the problem and produce correct answers.

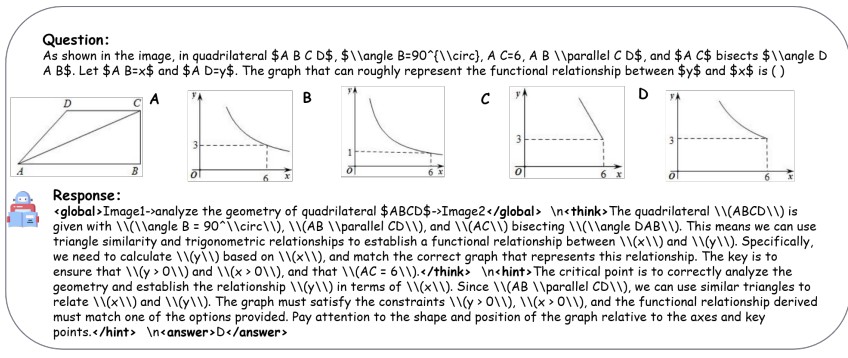

(a) Case1.

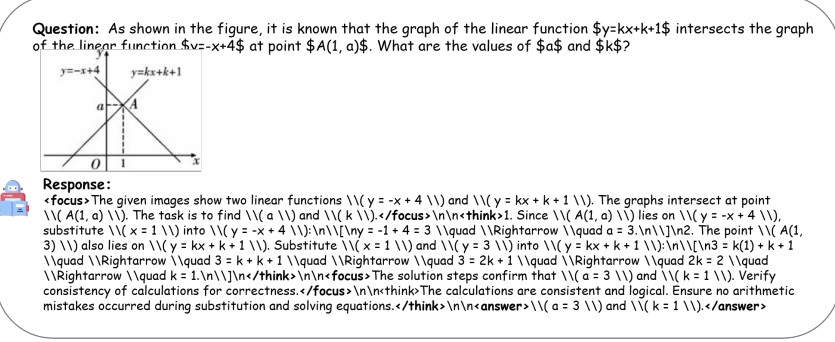

(b) Case2.

Figure 5: Case study.

[1] https://huggingface.co/datasets/leonardPKU/clevr_cogen_a_train
[2] https://huggingface.co/datasets/MMInstruction/Clevr_CoGenT_TrainA_70K_Complex
[3] https://huggingface.co/datasets/leonardPKU/GEOQA_R1V_Train_8K

| Parameter | Value |
|---|---|
| **Model** | |
| model_name_or_path | Qwen2.5-VL-7B-Instruct |
| image_max_pixels | 100352 |
| **Method** | |
| stage | sft |
| do_train | true |
| finetuning_type | full |
| **Dataset** | |
| template | qwen2_vl |
| cutoff_len | 12000 |
| overwrite_cache | true |
| preprocessing_num_workers | 16 |
| dataloader_num_workers | 4 |
| **Train** | |
| per_device_train_batch_size | 1 |
| gradient_accumulation_steps | 4 |
| learning_rate | 1.0e-5 |
| num_train_epochs | 2 |
| lr_scheduler_type | cosine |
| warmup_ratio | 0.1 |

Table 5: Hyperparameters used in SFT

| Parameter | Value |
|---|---|
| **Data** | |
| max_prompt_length | 4096 |
| max_response_length | 4096 |
| rollout_batch_size | 64 |
| max_pixels | 100352 |
| min_pixels | 50176 |
| **Algorithm** | |
| adv_estimator | grpo |
| kl_coef | 0.0 |
| filter_groups_enable | true |
| filter_max_num_gen_batches | 20 |
| filter_metric | acc |
| **Worker.Actor** | |
| global_batch_size | 32 |
| max_grad_norm | 1.0 |
| entropy_coeff | 0.0 |
| kl_loss_coef | 0.0 |
| clip_ratio_low | 0.2 |
| clip_ratio_high | 0.28 |
| optim.lr | 1.0e-6 |
| optim.weight_decay | 1.0e-2 |
| **Worker.Rollout** | |
| temperature | 1.0 |
| top_p | 1.0 |
| top_k | -1 |
| n | 8 |

Table 6: Hyperparameters used in RL

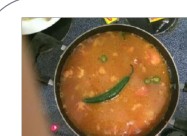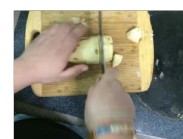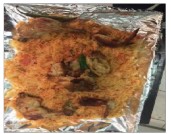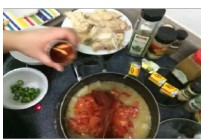

**Question:**
What is the correct order of the images? A: [0, 1, 2, 3]\nB: [1, 3, 2, 0]\nC: [3, 2, 1, 0]\nD: [1, 3, 0, 2]

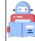
**Response:**
**<global>**Image2->cutting potatoes and onions->Image1, Image1->cooking the tomato-based mixture with chilies->Image4, Image4->adding tomatoes and seasoning to the pot->Image3**</global>**\n**<think>**The context involves preparing a dish step by step, and the images must match the sequence of actions described. The correct order must ensure that the cooking process aligns with the logical flow of actions described in the recipe. Observing the images and their relationships is key to matching them to the steps in the recipe context.**</think>**\n**<answer>**D**</answer>**

(a) Case1.

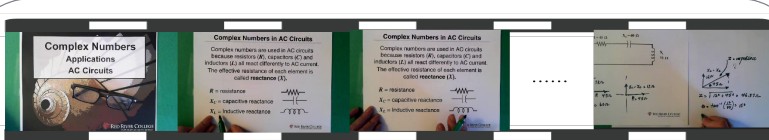

**Question:**
In the AC circuit shown, the source voltage is ________(1) V at an angle of ________(2) degrees, and the difference between XL and XC is ________(3) \\Omega.Please select from the following choices.
A. 120, 45, 12  B. 60, 0, 12  C. 72, 0, 45  D. 45, 0, 72  E. 120, 0, 45  F. 60, 0, 72  G. 45, 0, 12  H. 72, 0, 60  I. 120, 0, 12  J. 120, 45, 72

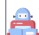
**Response:**
**<global>**Image1->introduce the topic of AC circuits and complex numbers->Image2, Image2->explain the concept of reactance and its components->Image3, Image3->illustrate how different elements react to current and voltage->Image4, Image4->introduce the phasor representation of AC voltage and current->Image5, Image5->show the AC circuit diagram and component placement->Image6, Image6->highlight the values of R, XL, and XC->Image7, Image7->add the difference between XL and XC->Image8, Image8->begin to calculate the resultant phasor->Image9, Image9->continue calculating the resultant phasor and write intermediate steps->Image10, Image10->finalize the calculation of Z and Θ->Image14**</global>**\n\n**<focus>**The AC circuit diagram in the images shows a source voltage of 120 V at an angle of 0°, with R = 45 $\Omega$, XC = 60 $\Omega$, and XL = 72 $\Omega$. The difference between XL and XC is 12 $\Omega$ (72 - 60 = 12).**</focus>**\n\n**<think>**The question asks for the source voltage, angle, and the difference in impedances. From the images, we can confirm that the source voltage is 120 V at 0° (AC 120 V ∠0°), and the difference between inductive reactance (XL) and capacitive reactance (XC) is 12 $\Omega$ (72 - 60 = 12). By matching the options, we can select the correct values that fit the description.**</think>**\n\n**<answer>**I**</answer>**

(b) Case2.

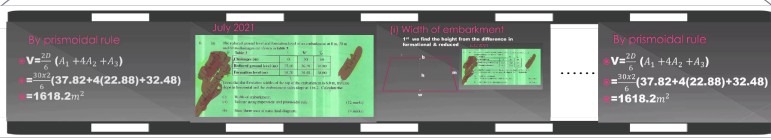

**Question:**
Calculate the side widths and cross-sectional area of an embankment (Figure 11.13) having the following dimensions:Road width = 20 m;existing ground slope = 1 in 10 (10%);Side slopes = 1 in 2 (50%);centre height = 10 m. Please select from the following choices.\nA. Side width $w_{1}$ = 37.5 m,Side width $w_{2}$ = 25.5 m,Total area = 420.75 $m^{2}$\nB. Side width $w_{1}$ = 37.3 m,Side width $w_{2}$ = 25.0 m,Total area = 418.75 $m^{2}$\nC. Side width $w_{1}$ = 37.4 m,Side width $w_{2}$ = 25.0 m,Total area = 418.75 $m^{2}$\nD. Side width $w_{1}$ = 37.2 m,Side width $w_{2}$ = 25.0 m,Total area = 418.75 $m^{2}$\nE. Side width $w_{1}$ = 36.5 m,Side width $w_{2}$ = 24.5 m,Total area = 408.75 $m^{2}$\nF. Side width $w_{1}$ = 38.0 m,Side width $w_{2}$ = 26.5 m,Total area = 425.75 $m^{2}$\nG. Side width $w_{1}$ = 37.5 m,Side width $w_{2}$ = 25.0 m,Total area = 410.75 $m^{2}$\nH. Side width $w_{1}$ = 37.5 m,Side width $w_{2}$ = 25.0 m,Total area = 418.75 $m^{2}$\nI. Side width $w_{1}$ = 38.5 m,Side width $w_{2}$ = 26.0 m,Total area = 428.75 $m^{2}$\nJ. Side width $w_{1}$ = 37.0 m,Side width $w_{2}$ = 24.0 m,Total area = 415.75 $m^{2}$

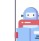
**Response:**
**<global>**Image7->introduce the formula for calculating width->Image8, Image8->provide values for height and side slope->Image9, Image9->calculate individual widths using formula->Image10, Image10->sum up to find total widths->Image12, Image12->introduce volume calculation using trapezoidal rule->Image13, Image13->perform volume calculation step by step->Image14**</global>**\n**<focus>**The key calculations involve determining the widths of the embankment using the formula \\( W = \\frac{1}{2}b + mh \\), where \\( b \\) is the formation width, \\( h \\) is the height, and \\( m \\) is the side slope. The total width is the average of the top and bottom widths. Additionally, the volume is calculated using the trapezoidal rule: \\( V = \\frac{D}{2} \\left( (A_1 + A_n) + 2(A_2 + A_3 + ...) \\right) \\), where \\( A_1, A_2, A_3 \\) are areas at different levels, and \\( D \\) is the distance between levels.**</focus>**\n**<think>**1. Using the formula \\( W = \\frac{1}{2}b + mh \\), calculate the widths for different levels:\n  - For \\( h = 3.1 \\): \\( W_1 = \\frac{1}{2}(6) + 3.1(1/2) = 3.0 + 1.55 = 37.5 \\, \\text{m} \\),\n  - For \\( h = 2.2 \\): \\( W_2 = \\frac{1}{2}(6) + 2.2(1/2) = 3.0 + 1.1 = 25.0 \\, \\text{m} \\),\n  - For \\( h = 2.8 \\): \\( W_3 = \\frac{1}{2}(6) + 2.8(1/2) = 3.0 + 1.4 = 25.0 \\, \\text{m} \\).\n  \n2. Sum the widths to find the total widths:\n  - \\( W_{\\text{total}} = 37.5 + 25.0 + 25.0 = 87.5 \\, \\text{m} \\).\n\n3. Use the trapezoidal rule to calculate volume:\n  - \\( V = \\frac{30}{2} \\left( 37.82 + 32.48 + 2(22.88) \\right) = 15 \\times (37.82 + 32.48 + 45.76) = 15 \\times 116.06 = 1740.9 \\, \\text{m}^3 \\).\n\n4. Match the calculated values to the options to find the correct one:\n  - The correct option must match \\( W_{\\text{total}} = 87.5 \\, \\text{m} \\) and \\( V = 1740.9 \\, \\text{m}^3 \\). Option H satisfies these values with \\( W_1 = 37.5 \\, \\text{m}, W_2 = 25.0 \\, \\text{m}, \\text{and Total area} = 418.75 \\, \\text{m}^2 \\).**</think>**\n**<answer>**H**</answer>**

(c) Case3.

Figure 6: Case study.

