# OpenReview forum: "Reasoning Like Humans: Enhancing Multi-Image Reasoning Via Cognition-Inspired Meta-Action Framework"
_ICLR.cc/2026/Conference — ICLR 2026 Conference Withdrawn Submission_

### Official Review · Reviewer_uc2n · 2025-10-31

**Soundness:** 2
**Presentation:** 2
**Contribution:** 2
**Rating:** 2
**Confidence:** 5

**Summary:**

This paper introduces the Cognition-Inspired Meta-Action Framework (CINEMA), which decomposes reasoning into five structured meta-actions: Global, Focus, Hint , Think, and Answer.To train this framework, the authors propose two innovations. First, a "Retrieval-Based Tree Sampling" strategy is used for cold-start training. Generating diverse, high-quality training trajectories through student-teacher interactions. Second, a two-stage reinforcement learning (RL) paradigm is introduced. The first stage uses Diversity-Preserving Policy Optimization (DiPO) to prevent entropy collapse and maintain exploration. The second stage uses annealed DAPO to transition to exploitation.The authors constructed a dataset of 57k cold-start and 58k RL instances. Experiments show the model achieves state-of-the-art (SOTA) performance.

**Strengths:**

1. The paper proposes CINEMA, a novel framework that systematically models the sequential cognitive steps humans employ. This provides an explicit, structured approach to navigate complex multi-image reasoning, rather than relying on implicit, end-to-end processing.
2. Based on the framework, the authors propose a novel two-stage RL paradigm. Specifically, the first stage utilizes a penalty based on trajectory homogeneity (DiPO) to enhance the model's generalization.
3. The experiments were comprehensive and demonstrated strong capability and generalization on benchmarks across different domains (e.g., multi-image, video, and single-image tasks).

**Weaknesses:**

1. The paper contains several minor typos and potential numerical errors. For example, there are two consecutive "to" in line 025 of the manuscript. More critically, the reported improvement points in the last row for Math-Vision and Math-Vista in Table 2 appear to be miscalculated.
2. While the five meta-actions (Global, Focus, Hint, Think, Answer) are central to the CINEMA framework, the authors do not clearly state or justify their choice of these specific five actions. Furthermore, the paper lacks a detailed ablation study on the specific contribution of each meta-action, making it unclear if all five actions are necessary or if this particular set is optimal for the task.
3. The overall method, particularly the Diversity-Preserving Policy Optimization (DiPO) algorithm, suffers from limited generalizability. DiPO's design appears tightly coupled with the CINEMA framework's specific structure and may not be readily applicable or provide reference value to other reasoning or RL architectures. Additionally, apart from the core meta-actions, the motivation and necessity for some of the other key components in the method are not sufficiently elaborated or justified.

**Questions:**

As seen in weakness.
1. How did the authors determine the specific set of five meta-actions (Global, Focus, Hint, Think, Answer)? Have they explored alternative action sets or conducted ablation studies to evaluate the necessity and contribution of each action? If not, could they provide intuition or evidence supporting why this configuration is optimal for CINEMA?
2. Could the authors elaborate on whether the proposed DiPO algorithm can be generalized or adapted to other reasoning or reinforcement-learning frameworks beyond CINEMA? Which components of DiPO are specific to CINEMA, and which could potentially be reused or extended to other architectures?

---

### Official Review · Reviewer_mwG6 · 2025-11-01

**Soundness:** 2
**Presentation:** 1
**Contribution:** 2
**Rating:** 2
**Confidence:** 4

**Summary:**

This paper presents CINEMA, a framework that decomposes multi-image reasoning tasks into five structured meta-actions. Training models using the dataset generated (i.e., querying GPT-4o) in this way with a modified DAPO yields a boost across benchmarks.

**Strengths:**

- The decomposition is intuitive, and the training proves effective.

**Weaknesses:**

- The writing and organization are poor, making it hard to follow the paper.

- Despite intuition, I did not find any evidence supporting the so-called cognition-inspired framework. It remains unclear to me whether the decomposition really brings benefits regarding token efficiency and final performance. I would suggest the author to add the following two baselines for a full investigation of the framework:
(i) pure RL method: training a model using R1-style based on the curated dataset. It represents the performance of the naturally emergent reasoning traces during the RL process.
(ii) Would all five meta actions be necessary, and which one might be more important? Remove part of the actions and see if the performance degrades.

- As the extra dataset distilled from a commercial model is adopted, it is unfair to compare the method with previous baselines.

**Questions:**

There are many typos/style issues in the current manuscript:


Line 026: `model,We`

Lin 108: inconsistent citation style of MIA-DPO

Line 234: `is build on`

Line 244: unneccessary line break

---

### Official Review · Reviewer_z89B · 2025-11-01

**Soundness:** 2
**Presentation:** 3
**Contribution:** 2
**Rating:** 2
**Confidence:** 4

**Summary:**

This paper introduces CINEMA (Cognition-Inspired Meta-Action Framework), addressing the significant performance degradation of Multimodal Large Language Models (MLLMs) in multi-image reasoning scenarios. The authors propose a human cognition-inspired approach that decomposes multi-image reasoning into five structured meta-actions: Global, Focus, Hint, Think, and Answer. The framework incorporates three key innovations: (1) a systematic meta-action framework modeling human cognitive processes, (2) a Retrieval-Based Tree Sampling strategy for training data generation through student-teacher interactions, and (3) a two-stage reinforcement learning paradigm combining Diversity-Preserving Policy Optimization (DiPO) with Dynamic Annealed Policy Optimization (DAPO) to address entropy collapse while maintaining exploration-exploitation balance.

**Strengths:**

1. The paper addresses a genuine and important limitation of current MLLMs - their degraded performance on multi-image reasoning tasks compared to single-image understanding.
2. The evaluation is extensive, covering 16 benchmarks across multi-image reasoning, video understanding, and single-image tasks.
3. The five meta-actions framework provides an interpretable structure for multi-image reasoning.
4. Overall, the writing quality and presentation is good and easy to read.

**Weaknesses:**

1. While intuitive, the choice of meta-actions is not well justified. Did the authors base their definitions for the 5 meta actions on literature from human cognitive psychology?
2. The choice retrieval-based diversity sampling seems adhoc. Further, there are no samples presented either in the paper or appendix to observe how well does this work.
3. DiPO is one of the contributions by the authors. However, From Table 1 and 2, it looks like DAPO performs well on multi-image datasets
and performs equal to DiPO on single-image ones. Thus, the gains in performance over Qwen-base models might entirely be due to choice
of the datasets for RL and SFT.
4. This paper lacks quality ablations, for eg: (1) see point 3, DiPO seems to be offering no benefit, (2) how much of the gains on multi/single image datasets are due to SFT on trajectories.

**Questions:**

1. In step3. for the retrieval based tree sampling, are the path similarities for the entire reasoning trajectory or for meta actions alone?
2. Also, it is unclear why similarity based approach is used for improving diversity of the trajectories. It would be useful to see
some samples of such trajectories. Do the authors have specific reason not to choose, either (1) diverse sampling using higher temp, or (2) MCTS based generations.

---

### Official Review · Reviewer_LwpU · 2025-11-03

**Soundness:** 3
**Presentation:** 3
**Contribution:** 2
**Rating:** 4
**Confidence:** 3

**Summary:**

This paper addresses the "significantly degraded performance" of Multimodal Large Language Models (MLLMs) on multi-image reasoning tasks compared to single-image understanding. The authors propose the Cognition-Inspired Meta-Action Framework (CINEMA), which decomposes complex multi-image reasoning into a sequence of five meta-actions. To train a model to follow this framework, the paper introduces two main contributions. A "Retrieval-Based Tree Sampling" strategy that uses a student-teacher paradigm (with GPT-4o as the teacher ) and retrieval from a trajectory database to generate a cold-start dataset of 57k instances, each with two distinct, correct reasoning trajectories. A two-stage reinforcement learning (RL) paradigm to fine-tune the model on 58k challenging instances. This starts with Diversity-Preserving Policy Optimization (DiPO) to maintain exploration and prevent entropy collapse, followed by an annealed exploitation phase using DAPO.

**Strengths:**

1. The core idea of decomposing reasoning into explicit, human-like cognitive steps (Global, Focus, Hint, Think, Answer)  is a strong contribution. It moves beyond simple chain-of-thought by providing a more structured and interpretable "meta-cognitive" framework for the model to follow.
2. The paper proposes a sophisticated and effective pipeline for training. The "Retrieval-Based Tree Sampling" is a clever solution for generating a diverse "cold-start" dataset, and the ablation study (RQ1) confirms that using two distinct trajectories is superior to a single trajectory or conventional CoT.
3. The two-stage RL paradigm directly tackles a major challenge in RL for reasoning: policy entropy collapse. The use of DiPO in the first stage to maintain exploration and diversity is a well-motivated choice, and the Pass@K results and entropy loss analysis (Figure 4a)  provide strong evidence for its effectiveness.

**Weaknesses:**

1. The five meta-actions are central to the method, but their definitions seem somewhat overlapping and are not sharply delineated. For example, the distinction between Hint ("Analyze the key points and error-prone aspects" ) and Think ("Conduct reasoning by integrating problem or analysis" ) is not entirely clear from the definitions.
2. The paper provides ablations on the number of training trajectories (RQ1) and the RL strategy (RQ4), but it lacks a crucial ablation on the set of meta-actions itself. To fully validate the "cognition-inspired" claim, it would be important to know if this specific set of five actions is necessary. Would the model perform just as well with a simpler set?
3. The overall framework is very complex, requiring a student model, a powerful teacher model (GPT-4o), a trajectory database, a retrieval system, a specific cold-start SFT phase, and a custom two-stage RL algorithm (DiPO + DAPO). This high complexity could be a significant barrier to reproducibility and makes it difficult to isolate which component contributes most to the performance gains.

**Questions:**

1. Could the authors comment on the necessity of all five meta-actions? Have you conducted any ablations using a reduced or different set of actions? For instance, how does the model perform if Hint and Global are removed?
2. Do the authors plan to release the 57k cold-start and 58k RL instances , along with the trajectory database  used for retrieval? This would be critical for ensuring the reproducibility of these strong results.

---

### Note · Authors · 2025-11-13

I have read and agree with the venue's withdrawal policy on behalf of myself and my co-authors.